# A combined high-throughput and high-content platform for unified on-chip synthesis, characterization and biological screening

Maximilian Benz [1], Arndt Asperger[2], Meike Hamester[2], Alexander Welle [3], Stefan Heissler[3] & Pavel A. Levkin [1,4✉]

Acceleration and unification of drug discovery is important to reduce the effort and cost of new drug development. Diverse chemical and biological conditions, specialized infrastructure and incompatibility between existing analytical methods with high-throughput, nanoliter scale chemistry make the whole drug discovery process lengthy and expensive. Here, we demonstrate a chemBIOS platform combining on-chip chemical synthesis, characterization and biological screening. We developed a dendrimer-based surface patterning that enables the generation of high-density nanodroplet arrays for both organic and aqueous liquids. Each droplet (among > 50,000 droplets per plate) functions as an individual, spatially separated nanovessel, that can be used for solution-based synthesis or analytical assays. An additional indium-tin oxide coating enables ultra-fast on-chip detection down to the attomole per droplet by matrix-assisted laser desorption/ionization mass spectrometry. The excellent optical properties of the chemBIOS platform allow for on-chip characterization and in-situ reaction monitoring in the ultraviolet, visible (on-chip UV-Vis spectroscopy and optical microscopy) and infrared (on-chip IR spectroscopy) regions. The platform is compatible with various cell-biological screenings, which opens new avenues in the fields of high-throughput synthesis and drug discovery.

[1] Karlsruhe Institute of Technology (KIT), Institute of Biological and Chemical Systems-Functional Molecular Systems (IBCS-FMS), Hermann-von-Helmholtz-Platz 1, 76344 Eggenstein-Leopoldshafen, Germany. [2] Bruker Daltonik GmbH, Fahrenheitstraße 4, 28359 Bremen, Germany. [3] Karlsruhe Institute of Technology (KIT), Institute of Functional Interfaces (IFG), Hermann-von-Helmholtz-Platz 1, 76344 Eggenstein-Leopoldshafen, Germany. [4] Karlsruhe Institute of Technology (KIT), Institute of Organic Chemistry (IOC), Kaiserstraße 12, 76131 Karlsruhe, Germany. ✉email: levkin@kit.edu

Drug discovery is a lengthy and highly expensive process. The development of a single new drug often takes over 20 years and costs on average \$2–4 billion[1–4]. Chemists in pharmaceutical and biotechnological companies synthesize thousands of compounds per year, requiring tons of expensive and valuable chemicals and consumables. Each individual compound must be characterized and screened for its biological activity. Automated high-throughput screening of compound libraries using microtiter plates is well-established in biology[5–7], although the physicochemical properties of polystyrene microtiter plates limits their use in chemical applications and specialized analytical techniques. Thus, a high-throughput approach is required for diverse, solution-based chemistry enabling simultaneous synthesis of hundreds to thousands of compounds. Miniaturization and parallelization of organic synthesis is crucial to enlarge the capacity of compound library design and reduce the effort, chemical consumption and associated costs. We previously reported the chemBIOS platform in a proof-of-principle study demonstrating on-chip miniaturized solution-based synthesis of 75 transfection agents and subsequent on-chip cell-biological screening in a process that was completed in only 3 days and required only 1 mL of solutions[8]. However, these improvements have been hindered because the existing analytical methods are not designed for nanoliter scale, on-chip high-throughput approaches. Liquid handling of nanoliter-sized volumes and sample transfer of thousands of compounds between synthesis, characterization, and screening platforms slows down the process of drug development and, thus, makes it expensive. There is a clear need to unify early-stage drug discovery on a multifunctional platform that enables sequential nano-scale synthesis of structurally diverse compounds, highly sensitive chemical characterization, and high-throughput biological screening on the same chip.

In this work, we develop a dendrimer-based surface-patterning method, evolved from the existing chemBIOS platform to expand its capabilities[8], that can be used to handle high-density nanodroplet arrays of both low (organic solvents) and high (aqueous solutions) surface-tension liquids, thus, enabling a broad range of chemical, analytical, and biological applications (chemBIOS) (Fig. 1). An indium–tin oxide (ITO) coating makes the platform conductive and therefore, compatible with on-chip high-throughput compound characterization by matrix-assisted laser desorption/ionization mass spectrometry (MALDI-TOF MS). We demonstrate highly sensitive on-chip approaches for infrared (IR) spectroscopy and high-content reaction monitoring by UV–Vis spectroscopy using this platform (Fig. 1). The open infrastructure and standardized format make chemBIOS adaptable for well-established assays and commercial devices and for both high-throughput (>50,000 droplets/plate) and high-content (>50,000 results/experiment) chemical, analytical, and biological screening (Fig. 1).

## Results

**Manufacturing and characterization of the platform.** Previously reported hydrophilic–hydrophobic functionalized droplet arrays used for high-throughput cell screenings were based on porous polymer substrates, making the surface oleophilic and incompatible with organic solvents with low surface tension[8–10]. Feng et al. developed a flat oleophilic–oleophobic patterned substrate enabling the formation of droplet arrays of organic solvents used for miniaturized and parallelized solution-based organic synthesis[8,11]. However, the low contrast between the advancing water contact angle of the oleophobic parts ($\theta_{adv} = 110.5 \pm 1.2°$) and the receding water contact angle of the oleophilic parts ($\theta_{rec} = 33.7 \pm 0.8°$) makes this substrate incompatible for handling high surface tension liquids[8]. To create arrays of droplets of both high and low surface tension liquids, the hydro-/oleophobic (omniphobic) patterns should have high advancing contact angles while the hydro-/oleophilic (omniphilic) parts should possess very low receding contact angles. We hypothesized that dendrimeric surface modification with high-density functional groups can increase the contrast of wettability properties leading to an omniphilic–omniphobic patterning. The dendrimeric surface modification is based on a poly(thioether) dendrimer synthesis developed by Killops et al. (Fig. 2a)[12]. The surface of a standard glass slide was silanized with triethoxyvinylsilane to produce a reactive, vinyl group-presenting surface. The dendrimeric layer was then synthesized in a repetitive two-step reaction cycle consisting of a photochemical thiol-ene click reaction with 1-thioglycerol followed by esterification with 4-pentenoic acid (Fig. 2b and Supplementary Fig. 1); three cycles produce a dendrimerized surface decorated with high-density alkene groups. This surface can then be photochemically functionalized either with 1-thioglycerol or 1$H$,1$H$,2$H$,2$H$-perfluorodecanethiol (PFDT), to yield an omniphilic or omniphobic surface, respectively. Omniphilic–omniphobic micropatterns of defined geometry can be created by sequential functionalization through a quartz photomask (Fig. 2c).

We assumed that an increase in functional group density should correlate with the wettability properties of the surface. To test this hypothesis, we compared the static water contact angle of different dendrimer generations (G0–G4) (Fig. 3a), starting with cysteamine hydrochloride functionalized surfaces[8,11]. In all, ± values of all contact angle measurements are standard deviations based on triplicate experiments. A slight increase in the static water contact angle (8.6°) was observed between the PFDT-modified G0 ($\theta_{stat}(H_2O)$ 104.3 ± 1.9°) and G4 ($\theta_{stat}(H_2O)$ 112.9 ± 1.5°) surfaces. A much greater decrease (23.8°) was observed between the cysteamine-modified G0 ($\theta_{stat}(H_2O)$ 55.9 ± 3.0°) and G4 ($\theta_{stat}(H_2O)$ 32.1 ± 5.7°) surfaces (Fig. 3a). There were no significant differences in $\theta_{stat}(H_2O)$ between the G2 and G4 surfaces. Therefore, we further analyzed the advancing and receding water contact angles of the G3 surfaces (Fig. 3b). In addition to omniphobic PFDT-modified G3 surfaces ($\theta_{adv}(H_2O)$ 124.9 ± 2.9°, $\theta_{stat}(H_2O)$ 116.6 ± 3.6°, $\theta_{rec}(H_2O)$ 111.2 ± 2.7°), we also investigated omniphilic cysteamine-modified surface ($\theta_{adv}(H_2O)$ 62.1 ± 3.0°, $\theta_{stat}(H_2O)$ 33.0 ± 3.0°, $\theta_{rec}(H_2O)$ 6.7 ± 2.9°), in which omniphilic nature was further improved by modifying the G3 surface with thioglycerol, leading to a surface with extremely low $\theta_{rec}(H_2O)$ ($\theta_{adv}(H_2O)$ 32.6 ± 2.2°, $\theta_{stat}(H_2O)$ 24.0 ± 0.3°, $\theta_{rec}(H_2O)$ 1.2 ± 0.6°).

A huge difference of 123.7° between the $\theta_{adv}(H_2O)$ of PFDT-modified G3 surface (124.9 ± 2.9°) and the $\theta_{rec}(H_2O)$ of thioglycerol-modified G3 areas ($\theta_{rec}(H_2O)$ 1.2 ± 0.6°) enabled the generation of stable nanodroplet arrays of high surface tension liquids, such as water ($\gamma_{lv}$ 72.8 mN m$^{-1}$) or cell suspensions. The difference between $\theta_{adv}$ of PFDT-modified and $\theta_{rec}$ of thioglycerol-modified G3 surface was above 80–100° even for organic solvents with a much lower surface tension (22.1–43.5 mN m$^{-1}$) than water (72.8 mN m$^{-1}$) enabling the generation of nanodroplet arrays of low surface tension liquids, including various common organic solvents such as ethanol ($\gamma_{lv} = 22.1$ mN m$^{-1}$), 1-decanol ($\gamma_{lv} = 28.5$ mN m$^{-1}$), dimethylformamide (DMF; $\gamma_{lv} = 37.1$ mN m$^{-1}$), and dimethyl sulfoxide (DMSO; $\gamma_{lv} = 43.5$ mN m$^{-1}$) on the same substrate, as well (Figs. 2d, 3c and Supplementary Table 1).

In the next step, thioglycerol-modified G0–3 surfaces were characterized by atomic force microscopy (AFM). In all, ± values of all AFM measurements are standard deviations based on triplicate experiments. An increase in surface roughness (Rq) of 90 ± 5 pm between each dendrimer generation was observed

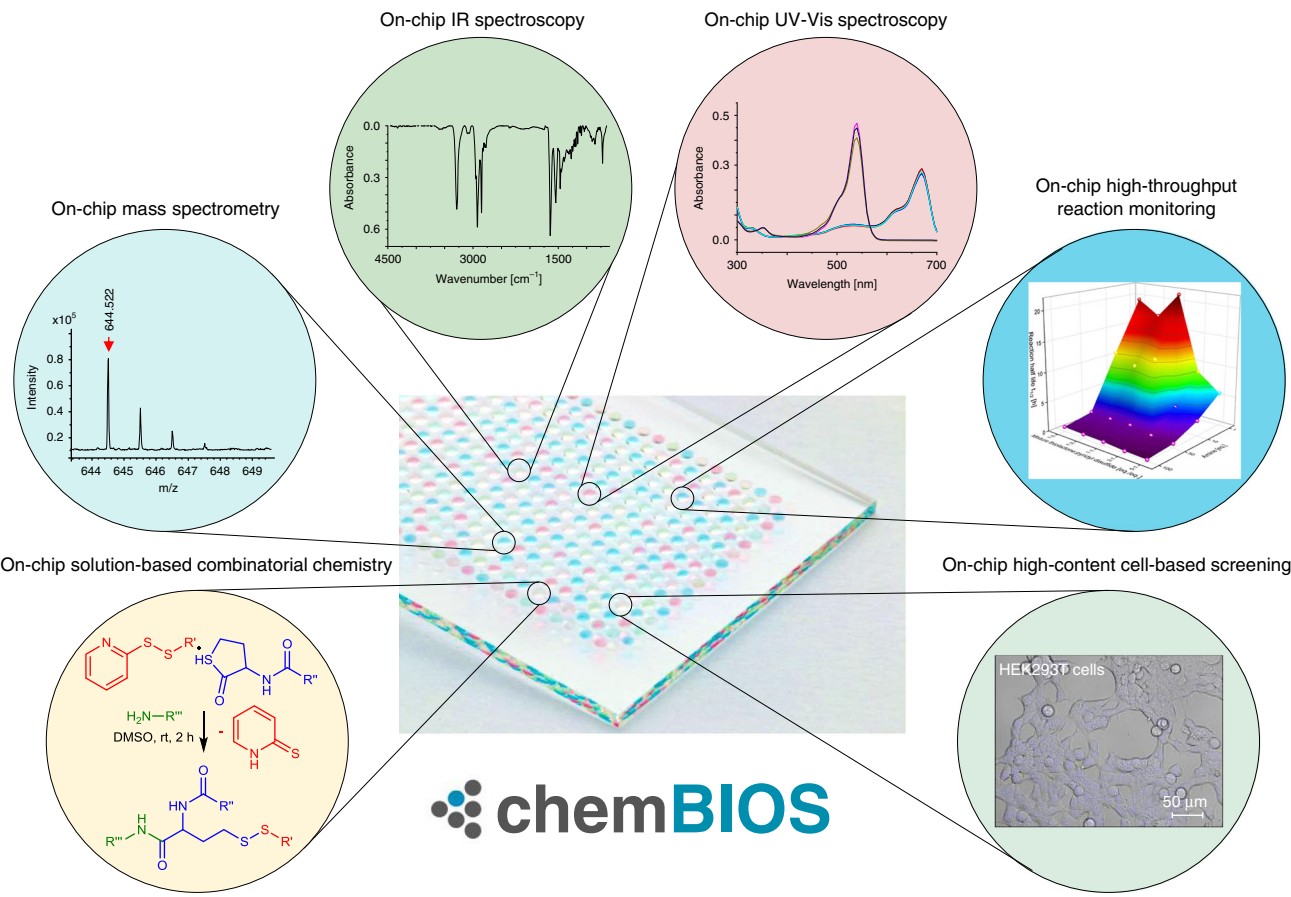

**Fig. 1 Chemical, analytical, and biological high-throughput methods compatible with the chemBIOS platform.** ChemBIOS enables on-chip solution-based synthesis, on-chip analytical characterization using various spectrometric and spectroscopic approaches for high-content screening and also unifies chemistry with biological high-throughput screening using nanodroplet arrays.

(Fig. 3d, Supplementary Fig. 2 and Supplementary Table 2). The thickness of each dendrimeric layer increased by $1.1 \pm 0.2$ nm per generation, which is consistent with a calculated maximum growth of 1.6 nm in height per dendrimer generation (Fig. 3e and Supplementary Tables 3 and 4). The increase in density of surface functional groups with dendrimer generation was confirmed using time-of-flight secondary ion MS (ToF-SIMS) (Supplementary Fig. 3). The decrease in the $Si^+$ signal arising from the silicone glass slide measured for each dendrimer generation matched the dendrimer growth (Supplementary Fig. 3). A well-defined chemical pattern was identified for the omniphilic–omniphobic-patterned G3 surface (Fig. 3f). The lateral resolution between omniphilic patterns and surrounding omniphobic borders was ~8 μm (Fig. 3g), enabling the generation of patterns with any defined geometry down to the low micrometer range (Fig. 3h) by photolithography using highly reactive dendrimer substrates.

**On-chip MALDI-TOF MS.** Recent developments in on-chip miniaturized and parallelized solution-based synthesis have enabled rapid synthesis of hundreds to thousands of compounds on a single plate, although on-chip characterization of large compound libraries remains a challenge[8]. On-chip high-throughput synthesis requires novel on-chip analytical methods that are compatible with low volume, low concentration, high-density arrays, and an on-chip platform. The goal of this research was to accelerate the drug development process by developing a combined platform for on-chip diverse chemical synthesis and straightforward compound characterization on the same plate.

MALDI-TOF MS enables characterization of thousands of compounds in seconds with high spatial resolution in the micrometer range using a conductive, flat, and arrayed plate format, which is therefore ideal for characterizing large compound libraries[13–16]. In order to enable MALDI-TOF MS on our chemBIOS platform, we applied developed surface-patterning method to a conductive ITO slide (Fig. 4a). Compounds can be applied or synthesized on-chip in an array format on the conductive slide followed by solvent evaporation, processing with matrix solution and characterization by MS (Fig. 4a). The slides remain transparent (Fig. 3i) and are therefore suitable for use in correlative analysis by optical methods such as microscopy (Supplementary Fig. 4), which is an advantage over standard MALDI stainless steel targets. All MALDI-TOF MS measurements of this study were performed on omniphilic–omniphobic patterned G3 ITO glass slides (Fig. 4a–c and Supplementary Fig. 4).

Miniaturizing chemical synthesis implies reduction of volumes and concentrations, respectively. Our first aim was to investigate the sensitivity of the on-chip MALDI-TOF approach to determine the limits of miniaturization of our chemBIOS platform. Three different lipid-like structures, lipidoids **1** (*m/z* 643.522), **2** (*m/z* 686.569), and **3** (*m/z* 672.553), dissolved in 2-propanol were dispensed to round spots (diameter 2.83 mm; border width: 1.67 mm) to yield a final amount of substance per spot in the range of 0.05–1000 fmol (Supplementary Fig. 4). The samples were co-crystallized with α-cyano-4-hydroxycinnamic acid (CHCA) matrix (Fig. 4a). All three lipidoids were identified with a sensitivity of 100 fmol substance per spot and an average S/N ratio of $38 \pm 10$ (Fig. 4c and Supplementary Table 5). The

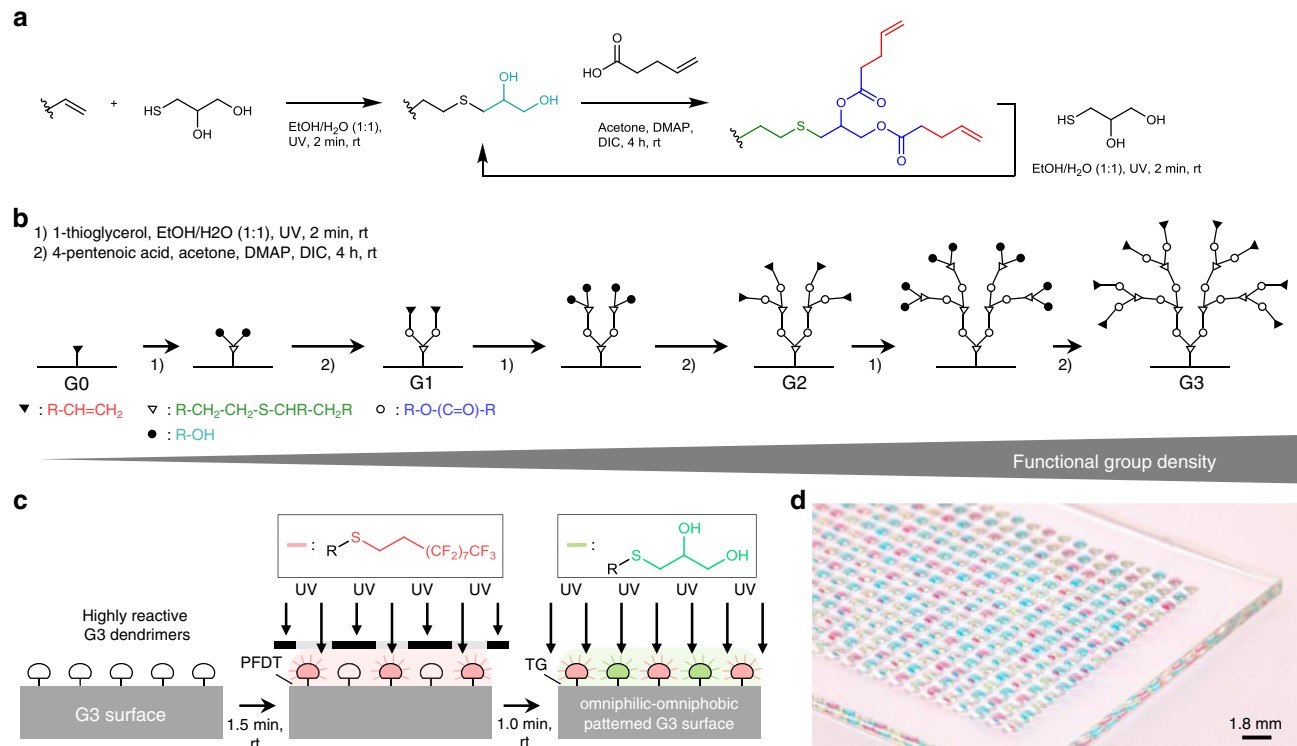

**Fig. 2 Manufacturing of patterned dendrimer slides based on grafted poly(thioether) dendrimers. a** The manufacture of dendrimeric slides begins with silanization of a glass slide with triethoxyvinylsilane, followed by a repetitive cycle involving a two-step reaction. An alkene reacts in a photochemical thiol-ene click reaction with 1-thioglycerol followed by an esterification with 4-pentenoic acid. **b** The amount of functional end-groups increases by $2^n$ ($n$: amount of repetitions) for each dendrimer generation (G) with reaching 8 reactive groups for the G3 surface. **c** A highly reactive, alkene-presenting dendrimer surface is patterned using the photochemical thiol-ene reaction. Omniphobic borders (red) are generated by a reaction with perfluorodecanethiol (PFDT), followed by the formation of omniphilic spots (green) by 1-thioglycerol (TG). **d** Photograph of a nanodroplet array of different compounds dissolved in dimethyl sulfoxide on an omniphilic–omniphobic patterned slide. Spot diameter: 900 μm; omniphilic border width: 225 μm; droplet volume: 100 nL; scale bar: 1.8 mm.

sensitivity was enhanced by a factor of four (S/N(100 fmol) 148 ± 25) by additional on-target washing with the matrix buffer solution (Fig. 4c, Supplementary Figs. 5 and 6, and Supplementary Table 5). In all, ± values of all MALDI-TOF MS measurements are standard deviations based on triplicate experiments.

Our next aim was to reduce the spot size, thereby increasing the throughput, and to evaluate the effect of the spot size on the sensitivity of the on-chip MALDI-TOF MS method. Solutions of lipidoids **1**, **2**, and **3** were dispensed to spots with diameters of 900 and 500 μm (border widths: 225 and 250 μm, respectively) to yield an amount of substance between 1000 and 0.05 fmol per spot. An equivalent amount of CHCA matrix solution was added to all samples. Miniaturization facilitated testing all three compounds on the same slide without altering the sample preparation parameters (Supplementary Fig. 4). This improves the ease and comparability of experiments, reduces the cost of materials, and increases the throughput, thus, enhancing work-flow efficiency. The limit of detection was 0.1 fmol for each lipidoid, demonstrating a 100-fold increase in the sensitivity of this method (Fig. 4b and c, Supplementary Fig. 6 and Supplementary Table 7). The average S/N ratio for all 10 fmol samples on 900 μm spots was 117 ± 37, which was 11 times higher than that of the 2.83 mm spots (S/N: 10.3 ± 0.5) (Fig. 4c). However, further reduction of the spot diameter to 500 μm did not yield a significant gain in MALDI-TOF sensitivity (S/N(10 fmol) 208 ± 92) compared to that for 900 μm spots (Fig. 4c). Further optimization of critically important parameters (aliquot size, concentration, and matrix solvent composition) might

improve the sensitivity for 500 μm spots. Nevertheless, a sensitivity limit of ~100 fmol allows detection of approximately only 60 million molecules on a single spot. Compared with standard chemical synthesis in flasks, where reactions are performed in the mol range (~$0.6 \times 10^{24}$ molecules) in milliliter-sized volumes, the chemBIOS platform enables not only practicable handling but also analysis of ultra-small amounts of substances (attomole range) and volumes (picoliters to nanoliters). Development of this ultra-sensitive chemBIOS plat-form for miniaturized and parallelized organic synthesis with integrated characterization is an important step toward high-throughput and high-content synthetic and analytical chemistry.

The dendrimeric surface modification and subsequent omniphilic–omniphobic patterning is not limited to silicon oxide or ITO-coated surfaces. Each hydroxy-presenting surface can be functionalized. Thus, we modified and patterned microtiter plate-sized MALDI stainless steel plates to create a $280 \times 180$ droplet array ($330 \times 330$ μm² spots separated by 60 μm borders) enabling the application and characterization of 50,400 compounds on a single MALDI plate (Supplementary Fig. 8). This demonstrates the potential for single plate ultra-high-throughput characterization. A broad variety of substance classes could be analyzed on the dendrimer-based omniphilic–omniphobic patterned stainless steel plates including peptides, proteins, carbohydrates, and small molecule drugs (Supplementary Fig. 8).

**On-chip IR spectroscopy.** ITO coating shows high transparency (>80% in the visible light region) (Fig. 3i) and high reflectance (~90% in the far-IR region)[17,18], properties which are used in

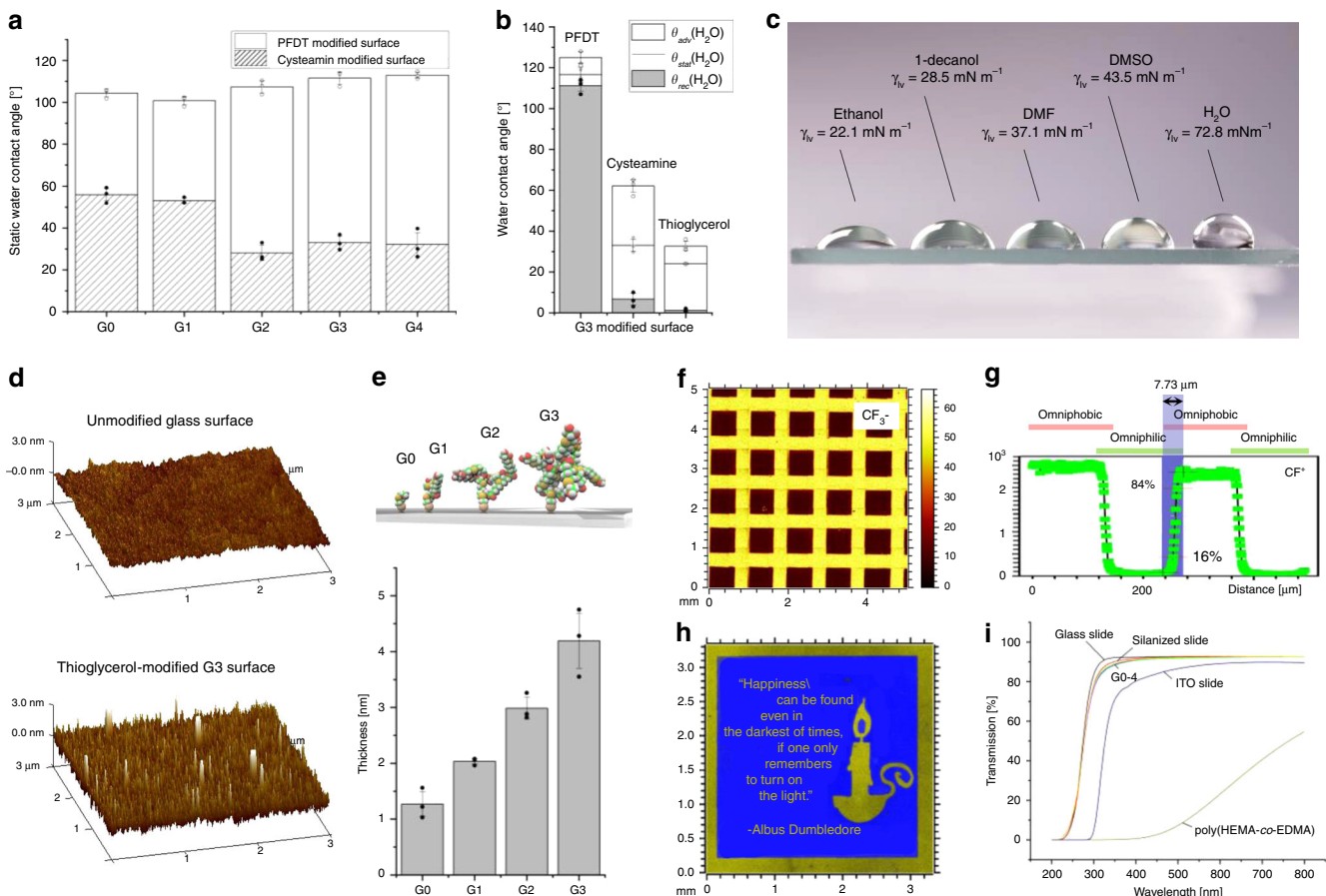

**Fig. 3 Characterization of dendrimer-grafted surfaces. a** Static water contact angles of PFDT-modified (omniphobic) and cysteamine-modified (omniphilic) G0–4 surfaces. **b** Advancing, static and receding water contact angles of G3 surface with different functionalizations (PFDT, cysteamine and thioglycerol). **c** Photograph of droplets of various low and high surface tension liquids on an omniphilic–omniphobic-patterned G3 surface. From left to right: ethanol ($\gamma_{lv}$ = 22.1 mN m$^{-1}$), 1-decanol ($\gamma_{lv}$ = 28.5 mN m$^{-1}$), dimethyl formamide ($\gamma_{lv}$ = 37.1 mN m$^{-1}$), dimethyl sulfoxide ($\gamma_{lv}$ = 43.5 mN m$^{-1}$), and water ($\gamma_{lv}$ = 72.8 mN m$^{-1}$). **d** AFM images of an unmodified glass surface and a thioglycerol-modified G3 surface. **e** Schematic visualization of thioglycerol-modified G0–3 surfaces and corresponding coating thicknesses measured by AFM. **f** ToF-SIMS analysis of lateral distribution of CF$_3^-$ signals showing PFDT–thioglycerol—omniphilic–omniphobic—surface patterning. Spot size: 700 × 700 µm$^2$. Color scale: [counts]. **g** CF$^+$ line scan perpendicular to a PFDT stripe pattern having 250 µm pitch, including indication of patterning fidelity by step analysis applying the 84/16% criterion. Lateral resolution of ~8 µm between PFDT-modified and thioglycerol-modified G3 areas. **h** ToF-SIMS stage scan of a customized pattern. Blue: CF$_3$; yellow: Sum of CH$_3$O, C$_2$H$_5$O, and C$_4$H$_5$O. **i** UV–Vis transmission spectra showing optical properties of thioglycerol-modified G0–4 glass surfaces, unmodified glass slide, ITO slide, and poly (HEMA-co-EDMA) surface. Data represent mean ± standard deviation based on triplicate experiments; $n = 3$ independent experiments. Source data are provided as a Source Data file.

biology for conventional light microscopy and IR microscopy to yield additional chemical information[19,20]. We aimed to develop an on-chip IR spectroscopy approach that enables mapping of compound libraries and thereby, obtain structural information of compounds in individual spots. An ITO-coated, omniphilic–omniphobic-patterned G3 glass slide was used as our analytical platform. Lipidoid **1** was applied at different concentrations to slides with round spots (diameter of 900 or 500 µm) (Fig. 4d, e). The lipidoid was identified down to 0.3 µg mm$^{-2}$ (94.7 fmol per 500 µm spot) (Fig. 4e). The IR spectrum for each compound in the array was used for structural analysis (Fig. 4f). Similar results were observed for spots with a diameter of 900 µm (Supplementary Fig. 9). On-chip IR imaging in an array format offers the potential for non-destructive and position-coded collection of chemical information and characterization of synthesized compound libraries. Although the sensitivity of on-chip IR spectroscopy is lower than that of MALDI-TOF MS, it remains a highly sensitive and complementary method for on-chip characterization and thus, an essential technique in miniaturized and parallelized chemistry. However, light-scattering between the

droplet–air interface limits this approach to measurements in the dried state. Further investigations are needed to develop methods for on-chip IR spectroscopy of liquids, which would be useful for on-chip in situ reaction monitoring, as well as a UV–Vis spectroscopy approach, which we developed in the next step.

**On-chip UV–Vis spectroscopy.** Many methods used for quantitative analysis of different reaction parameters are end-point driven and require numerous repetitions that are time-consuming and labor-intensive. Probe-based spectroscopy enables real-time reaction monitoring[21], although this requires specialized devices and is limited in terms of miniaturization and parallelization potential. Here we aimed to develop a workflow for on-chip solution-based synthetic chemistry that enables miniaturized and parallelized high-throughput experimentation as well as in situ reaction monitoring. We hypothesized that due to excellent optical properties, the chemBIOS platform should be compatible with on-chip UV–Vis monitoring. The transmittance of thioglycerol-modified G0–4 surfaces exceeded 90% in the visible region and 80% in the ultraviolet (UV-A/-B) region (Fig. 3i).

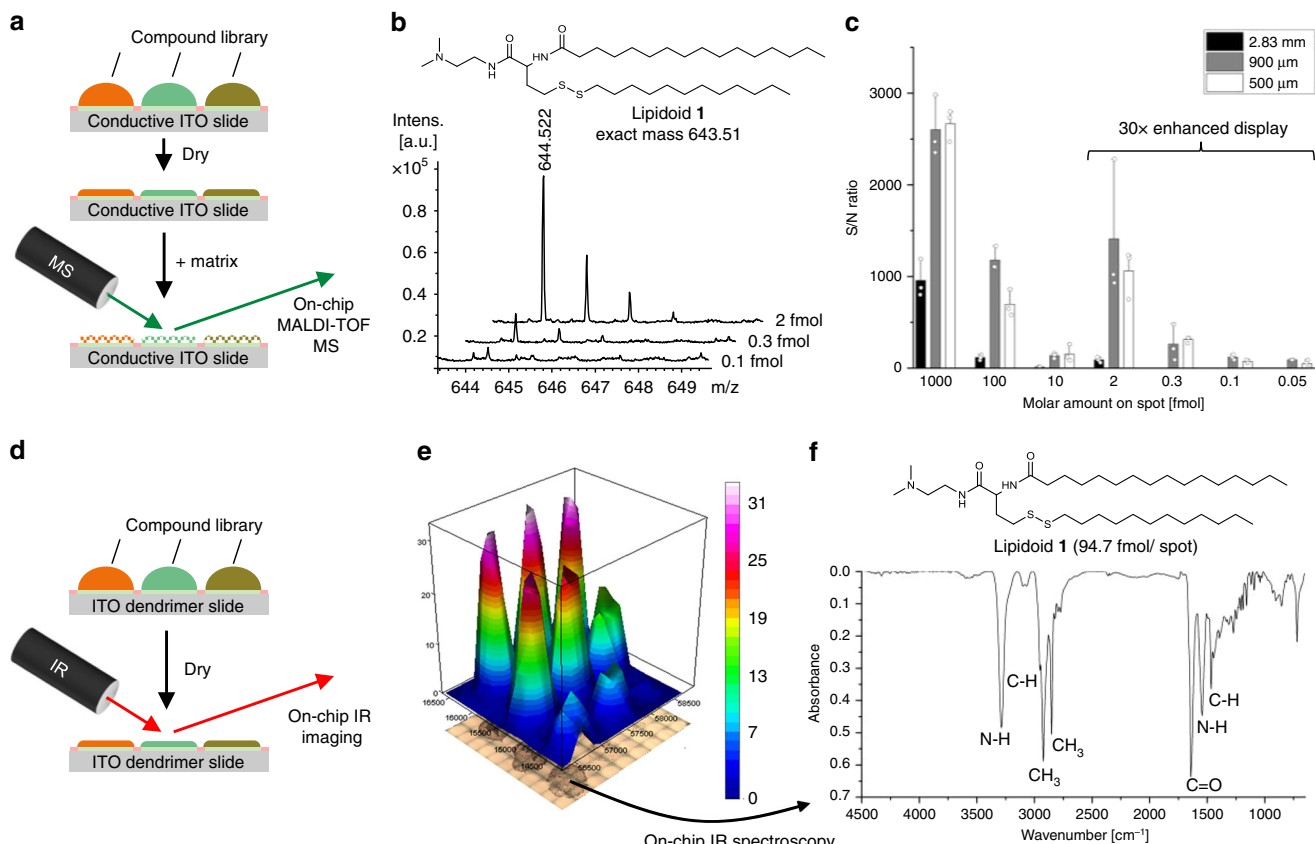

**Fig. 4 Indium-tin oxide substrates enable on-chip characterization by MALDI-TOF mass spectrometry and IR spectroscopy. a** Schematic showing the process of on-chip characterization by MALDI-TOF MS. A compound library generated on a conductive, dendrimer-modified, and patterned ITO slide can be processed and co-crystallized with matrix solution prior to MALDI-TOF analysis. **b** MALDI-TOF mass spectra of 2, 0.3, and 0.1 fmol per spot of lipidoid 1. Spot diameter: 500 μm; spot distance: 250 μm. The MALDI-TOF measurements were performed on patterned ITO glass slides. **c** Bar chart showing signal-to-noise (S/N) ratio obtained from on-chip MS analysis of lipidoid **1** in spots of different sizes. Data represent mean ± standard deviation based on triplicate experiments; $n = 3$ independent experiments. **d** Schematic diagram showing the process of on-chip characterization by IR spectroscopy. Non-invasive, on-chip characterization of a compound library by IR spectroscopy acquired after evaporation of the solvent. **e** On-chip IR imaging of several spots containing different amounts of lipidoid **1** per spot. Color scale: Absorbance [a.u.]. **f** IR spectrum of lipidoid **1** (94.7 fmol per spot). Spot diameter: 500 μm; spot distance: 250 μm. Source data are provided as a Source Data file.

We designed and 3D-printed a slide adapter that meets the standards of ANSI/SLAS 1-2004 through ANSI/SLAS 4-2004, which is therefore compatible with standard UV–Vis plate readers (Supplementary Fig. 10). Droplet arrays can be trapped between two sandwiched patterned slides within a fixed distance (1 mm), allowing analysis of individual droplets using a plate reader (Fig. 5a). In a proof-of-principle study, droplets of rhodamine 6 G and methylene blue were applied in a checker-board pattern on slide A, inserted into the slide adapter, and sandwiched and trapped with slide B before UV–Vis absorbance measurement. Spatially separated UV–Vis spectra were observed from each individual droplet (Fig. 5b).

Next, we applied our on-chip UV–Vis approach to investigate and optimize a one-pot synthesis of lipid-like structures[22]. The three-component reaction is initiated by an amine-based ring-opening of a thiolactone followed by a disulfide exchange with a pyridyl disulfide (Fig. 6a). The resulting UV-absorbing by-product, 2-thiopyridone ($\lambda_{max} = 365$ nm) was used as an indicator to monitor the reaction progress. Two compound libraries were prepared: (i) slide A contained four blocks of amine A1 (1, 10, 50, and 100 eq.) in triplicate, and (ii) slide B contained mixtures with different ratios of thiolactone T14 to pyridyl disulfide PY12 (1:5, 1:3, 1:1, 3:1, and 5:1; eq./eq.) applied in rows perpendicular to the blocks of slide A (Fig. 6b). Both slides were sandwiched in the slide adapter, thus, simultaneously initiating all

reactions under equal environmental conditions (Fig. 6c). All reactions were carried out at room temperature and monitored by UV–Vis spectrometry at 5-min intervals over a period of 70 h. Twenty different reaction parameters, each in triplicates, were analyzed resulting in 50,400 data points within a single experiment (Fig. 6d). The reaction half-life ($t_{1/2}$) decreased as the thiolactone component ratio increased and decreased more rapidly as the amine concentration increased. This indicated that the first step of the reaction (ring-opening of the thiolactone by the amine) is rapid and the following disulfide exchange is rate-determining in the one-pot synthesis.

**Biological compatibility.** The ultimate goal of accelerating the screening of novel compound libraries and increasing throughput is to unify chemical processes, including compound library synthesis, characterization, and biological high-throughput screening[8,23]. However, the lack of compatibility between the diverse platforms and infrastructures (flasks, tubes, bottles, polystyrene plates, plastic/glass pipettes, syringes, filters, etc.) of these methods with different material properties (chemical resistance vs. biological compatibility) limits the applicability of high-throughput approaches and slows drug and biotechnological development. Besides compatibility with a variety of organic and aqueous solutions, the platform must be cell compatible. The

**a**

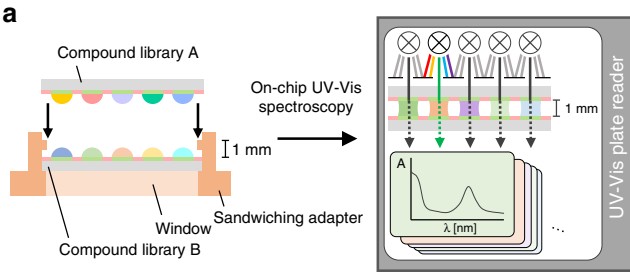

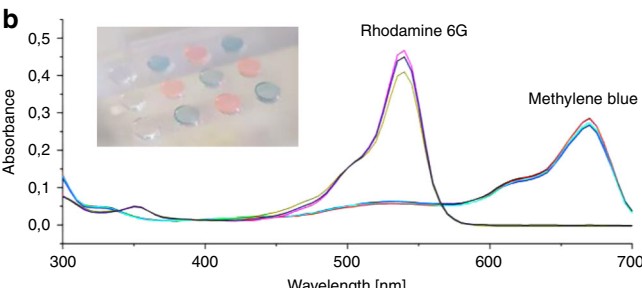

**Fig. 5 On-chip characterization by UV–Vis spectroscopy. a** Schematic diagram showing the process of on-chip UV–Vis spectroscopy. Droplets can be trapped between two slides sandwiched within a distance of 1 mm in a 3D-printed adapter. On-chip UV–Vis absorbance can be measured using commercial plate readers. **b** On-chip measured UV–Vis spectra of rhodamine 6 G and methylene blue applied on-chip in a checkerboard pattern and trapped between two dendrimer slides. Source data are provided as a Source Data file.

viability of three cell lines (HeLa, HEK293T, and Jurkat) dispensed to each spot of a thioglycerol-modified G3 slide and cultured for 24 h was evaluated by fluorescence microscopy after life/death staining and by morphology (Supplementary Fig. 11). The viability of each cell line exceeded 97%, demonstrating the compatibility of this platform for cell culturing and opened the possibility for further biological screenings in follow-up studies.

In summary, we developed a covalent surface modification based on a surface-grafted poly(thioether) dendrimer resulting in an increase in the density of surface presentation of reactive alkene groups (Fig. 2a, b). Further photochemical patterning of this surface with 1-thioglycerol and PFDT sequentially through a photomask resulted in the formation of omniphilic spots surrounded by omniphobic borders (Fig. 2c). The high contrast of advancing and receding contact angles of both aqueous and organic liquids between the omniphilic and omniphobic patterns, presenting high contrast in surface wettability, enabled the formation of nanodroplet arrays (1–3000 nL depending on the spot size) of both low surface tension (organic) and high surface tension (aqueous) liquids (Figs. 2d and 3b, c). This offers the potential for use in a broad range of chemical, analytical, and biological applications, such as miniaturized and parallelized on-chip solution-based synthesis or on-chip biological screening (chemBIOS) (Fig. 1). Each nanodroplet forms a distinct vessel that can be used individually. Chemicals, agents, or cell suspension can be added at any time during an assay or whole compound libraries can be combined within a single step (Fig. 5a). The dendrimer-based omniphilic–omniphobic platform enabled generation of nanodroplet arrays of both low-surface tension liquids (organic solvents) and high-surface tension liquids (aqueous solutions) on the same substrate. This opened the possibility to combine on-chip synthesis, on-chip characterization and on-chip biological screening in an iterative cyclic workflow (Supplementary Fig. 12). We aimed to study the sensitivity of various analytical methods and evaluated their compatibility for

on-chip compound characterization. An additional ITO coating made the chemBIOS surface conductive and, thus, compatible with MALDI-TOF MS (Fig. 4a). The sensitivity of the on-chip MALDI-TOF MS approach was investigated using an exemplar lipid-like compound class. An increased sensitivity down to the attomole range per spot was observed for spots of increasingly smaller size (diameter 900 and 500 μm) (Fig. 4b, c). The synergy between higher sensitivity for smaller spot sizes enable miniaturization and, thus, parallelization of the on-chip approach. We created a nanodroplet array in the microtiter plate size containing 50,400 droplets ($330 \times 330 \mu m^2$ spots; border width: 60 μm) demonstrating the possibility for ultra-high-throughput screening (Supplementary Fig. 8). Furthermore, the IR-reflective properties of the ITO coating enabled IR imaging of a compound array and on-chip IR spectroscopy of selected compound spots (Fig. 4e, f). The sensitivity of the IR spectroscopy approach was found to be 95 fmol/500 μm spot (Fig. 4f). The dendrimer-modified chemBIOS platform showed excellent transparency in the visible and UV region (Fig. 3i) and is therefore compatible with optical analysis methods, enabling on-chip high-throughput in situ reaction monitoring using UV–Vis spectroscopy (Figs. 5 and 6). An on-chip, solution-based, combinatorial three-component reaction was carried out to demonstrate the possibility of an on-chip chemical reaction optimization with on-line reaction progress monitoring. Thus, we simultaneously screened 60 reactions resulting in the generation of 50,400 data points within a single experiment (Fig. 6). The results provided insights into the reaction mechanism that could be used for reaction optimization. The dendrimer-based surface modification combines compartmentalization, excellent optical properties (UV–Vis transparent and IR reflective), chemical resistance (glass substrate compatible with organic solvents) and conductive material properties and, thus, makes it possible to combine important characterization methods (for chemical and biological readouts), such as on-chip highly sensitive MS, on-chip spectroscopy, and on-chip (optical or potentially electron) microscopy in one multifunctional platform. ChemBIOS is compatible with various adherent and suspension cell lines (HeLa, HEK293T, and Jurkat) and is therefore suitable for high-throughput cell-based screening (Supplementary Fig. 11). ChemBIOS combines on-chip high-throughput chemical synthesis, high-content reaction monitoring, highly sensitive compound characterization and biological screening and, thus, unifies all areas of early-stage drug discovery.

## Methods

**Preparation of dendrimeric-modified and patterned surfaces.** The surface of standard microscope glass slides ($25 \times 75 \times 1$ mm, width × length × thickness, Schott Nexterion) was cleaned for 10 min using an UVO-Cleaner 42-220 (Jelight) and then silanized in the gas phase using 400 μL triethoxyvinylsilane (Sigma-Aldrich) at 80 °C for 20 h. The slide was washed with acetone and ethanol and then dried by compressed air. The solid-phase synthesis of one dendrimer generation consisted of two steps: (i) 300 μL 10% (v/v) 1-thioglycerol (Sigma-Aldrich) in ethanol/water (1:1, v/v, Merck Millipore) containing 1 wt% 2,2-dimethoxy-2-phenylacetophenone (DMPA, Sigma–Aldrich) was applied to the alkene-presenting surface. A thiol-ene click reaction was carried out by irradiating the surface with 260 nm UV light (OAI model 30) at 3 mW cm$^{-2}$ intensity for 1.5 min. The slide was washed with ethanol and then dried by compressed air. (ii) Esterification of the hydroxy-presenting surface was performed by incubating the slides in esterification mixture (112 mg 4-(dimethylamino)pyridine (Novabiochem), 250 μL pentenoic acid (Sigma-Aldrich) and 180 μL N,N'-diisopropylcarbodiimine (Alfa Aesar) in 90 mL acetone (Merck Millipore)) for 4 h while shaking at room temperature. The surface was patterned by first applying 300 μL 10% (v/v) PFDT (Sigma-Aldrich) in acetone containing 1 wt% DMPA (Sigma-Aldrich). The thiol-ene photoclick reaction was carried out by irradiating the surface through a photomask (Rose Fotomasken) with 260 nm UV light (OAI model 30) at 3 mW cm$^{-2}$ intensity for 1 min to create omniphobic borders. The slide was washed with acetone and then dried by compressed air. Omniphilic spots were formed by applying 300 μL 10% (v/v) 1-thioglycerol (Sigma-Aldrich) in ethanol/water (1:1, v/v, Merck Millipore) containing 1 wt% DMPA (Sigma-Aldrich) onto the patterned surface and

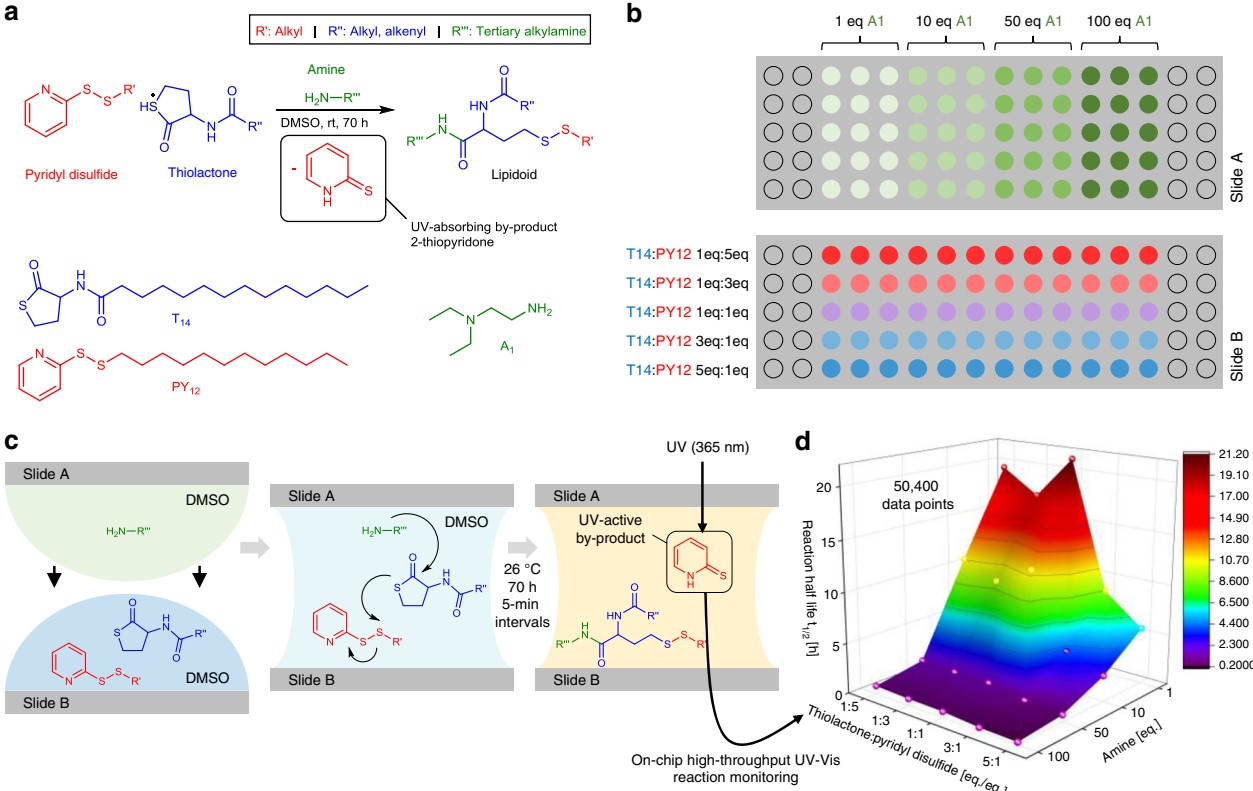

**Fig. 6 On-chip high-throughput reaction monitoring. a** Reaction scheme of the three-component lipidoid synthesis based on amine A1, thiolactone T14, and pyridyl disulfide PY12. **b** Array layout for parallel on-chip synthesis of sixty lipidoids. Amine A1 was dispensed in different concentrations column-by-column on slide A. Mixtures with different ratios of thiolactone T14 to pyridyl disulfide PY12 were dispensed row-by-row on slide B. All reactions were simultaneously initiated by sandwiching both slide. **c** Schematic visualization of the sandwiching process and resulting on-chip solution-based reaction. The UV-absorbing by-product 2-thiopyridone allowed on-chip in situ reaction monitoring by UV–Vis spectroscopy. **d** 3D plotting of the reaction half-life ($t_{1/2}$ for each reaction condition) demonstrates the power of such a high-throughput system to investigate reaction parameters and mechanistical aspects of the reaction. Source data are provided as a Source Data file.

irradiating the surface with 260 nm UV light (OAI model 30) at 3 mW cm$^{-2}$ intensity for 1.5 min. The slide was washed with ethanol and then dried by compressed air.

Dendrimeric modification and omniphilic–omniphobic patterning of ITO-coated slides were carried out using the protocol described above using commercial ITO slides (Bruker). Modification and patterning of stainless steel HTS sample plates (Bruker) was conducted accordingly.

**Surface characterization.** Advancing, static, receding contact, and sliding angles were measured using a Drop Shape Analyzer DSA25 (Krüss). Solvents were purchased without further purification from Merck Millipore.

AFM was performed using a Dimension Icon AFM (Bruker) under the following conditions: tapping mode, scan rate 1 Hz, sample/line 512 and AFM Cantilevers HQ:NSC15/AI BS-15 (Innovative Solutions Bulgaria Ltd.). Thickness was measured by scratching the surface with a tweezer and analyzing the depth of the scratch compared to the intact dendrimer surface surrounding the scratch.

ToF-SIMS analysis was performed using a ToF-SIMS-5 instrument (ION-TOF GmbH) equipped with a Bi cluster primary ion source and a reflectron-type ToF analyzer. UHV base pressure during analysis was <2 × 10$^{-9}$ mbar. For high mass resolution the Bi source was operated in a bunched mode providing short Bi$_3^+$ primary ion pulses at 25 keV energy, a lateral resolution of ~4 μm, and a target current of 0.4 pA. The short pulse length of 1 ns allowed for high mass resolution (8500 m/Δm). For large fields of view (Fig. 3f and h) the primary beam and the sample stage were rastered, For high lateral resolution (Fig. 3g) the primary gun was operated in a non-bunched mode combined with delayed extraction. The lateral resolution of the instrument is better than 0.5 μm in this case. For charge compensation on glass slide without ITO an electron flood gun (21 eV) was applied and the reflectron tuned accordingly. Primary ion dose was held below the static limit of 1 × 10$^{11}$ ions cm$^{-2}$, spectra were mass calibrated on the omnipresent signals C$^+$, CH$^+$, CH$_3^+$, C$_2$H$^+$ or C$^-$, CH$^-$, CH$_2^-$ C$_2^-$.

Transparency analysis was performed using a Lambda 35 UV–Vis spectrometer (PerkinElmer) equipped with a solid sample holder (PerkinElmer).

**On-chip MALDI-TOF MS.** Three lipidoids **1**, **2**, and **3** (≥5 replicates per slide) were tested on 2.83 mm, 900, and 500 μm spots of dendrimer-modified and patterned ITO slides by MALDI-TOF MS. Using a non-contact liquid dispenser (I-DOT, Dispendix) at a dispensing pressure set to 50–75 LA, sample solution was dispensed as follows: 500 nL per 2.83 mm spot, 50 nL per 900 μm spot, and 31 nL per 500 μm spot. Therefore, stock solutions of 1 μmol mL$^{-1}$ of all three samples in 2-propanol (Merck Millipore) were prepared. Aliquots were prepared out of this stock solution to reach an absolute amount of substance per spot of 1000, 100, 10, 2, 0.3, 0.1, and 0.05 fmol. The solvent was evaporated at room conditions after dispensing. Subsequently, 500 nL 5 mg mL$^{-1}$ CHCA (Alfa Aesar) dissolved in 70% 2-propanol with 0.1% trifluoroacetic acid (Sigma-Aldrich) and 1 mM ammonium dihydrogen phosphate (Merck Millipore) as additives were dispensed to each sample of a 2.83 mm spot, resulting in a matrix concentration of 397 ng mm$^{-2}$. Similarly, 50 nL (resulting in 391 ng mm$^{-2}$) and 31 nL (resulting in 388 ng mm$^{-2}$) of a 2.5 mg mL$^{-1}$ CHCA solution with the same solvent and additive conditions were printed onto each sample of a 900 and 500 μm spot. The solvent was dried at room conditions. Measurements were performed in reflector positive operation mode using a Bruker rapifleX MALDI-TOF/TOF system. The laser repetition rate was 10,000 Hz and 10,000 laser shots were accumulated per spectrum from 50 different raster positions within a sample spot. Spectra were acquired from three replicate spots for each sample dilution level. Peak areas/intensities and S/N ratios were determined using a centroid peak finder. S/N ratios of each amount of substance per area unit were determined and the limit of detection was estimated by comparing the sample spectrum with a blank matrix spectrum as a reference.

An additional on-target washing of prepared spots was carried out by applying 2 μL washing buffer (1 mM NH$_4$H$_2$PO$_4$, 0.1% TFA in water) to each 2.83 mm spot. The solution was removed after an incubation time of 1 s. On-target washing of slides with 900 and 500 μm spots as performed by immersing the slide into a washing buffer reservoir for 1 s. The slide was then dried at room temperature.

MALDI-TOF spectra of further classes of test substances measured on a patterned stainless-steel plate covering a wide range of molecular size (peptides, proteins, carbohydrates, small drug molecules) were acquired applying the following experimental conditions:

*Bovine serum albumin tryptic digest*: MALDI matrix HCCA; reflector positive operation mode; 6000 laser shots accumulated.

*Myoglobin (horse)*: MALDI matrix 2,5-DHAP; linear positive operation mode; 5000 laser shots accumulated.

*Maltoheptaose*: MALDI matrix 2,5-DHB/1 mM NaCl; reflector positive operation mode; 4000 laser shots accumulated.

*Verapamil*: MALDI matrix HCCA; reflector positive operation mode, 4000 laser shots accumulated.

**On-chip IR spectroscopy**. Aliquots of lipidoid **1** (40.4 mg mL$^{-1}$) dissolved in toluene (Merck Millipore) were dispensed to dendrimer-modified and patterned ITO slides (30 nL per 500 µm spot; 100 nL per 900 µm spot) using a non-contact liquid dispenser (I-DOT, Dispendix) at the following final concentrations: 6.3, 3.2, 0.7, and 0.3 µg mm$^{-2}$ per 900 µm spot (dispensed volume: 100 nL), and 6.1, 4.2, 3.0, 2.4, 1.8, 1.2, 0.6, 0.3, and 0.06 µg mm$^{-2}$ per 500 µm spot (dispensed volume: 30 nL). The solvent was allowed to dry at room conditions. Fourier-transform IR microscopy and spectroscopy were performed using a Bruker HYPERION 3000 microscope equipped with a ×15 IR objective. The spectral range of 4500–650 cm$^{-1}$ was recorded with a spectral resolution of 4 cm$^{-1}$ (32 scans per spot) using gold substrate as a reference. Measurements were performed in reflectance mode and converted in absorbance values.

**On-chip UV–Vis spectroscopy**. Aliquots (5 µL) of 36 µM methylene blue (Sigma-Aldrich) and 40 µM rhodamine 6 G (Sigma-Aldrich) dissolved in DMSO were dispensed to a G3 dendrimer slide (spot diameter: 2.83 mm, spot pitch: 1.67 mm) in a checkerboard pattern using a non-contact liquid dispenser (I-DOT, Dispendix). The slide was sandwiched with another G3 dendrimer slide in a 3D-printed sandwiching adapter with a distance of 1 mm between both slides. The UV–Vis absorbance spectrum of each spot was measured by a BioTek Synergy H1 plate reader within a spectral range of 300–700 nm and a spectral resolution of 5 nm.

**On-chip UV–Vis high-content screening**. The following stock solutions were prepared: thiolactone T14 in DMSO (Sigma-Aldrich) at 1.31, 3.93, and 6.55 mg mL$^{-1}$; pyridyl disulfide PY12 in DMSO at 1.25, 3.74, and 6.23 mg mL$^{-1}$; amine A1 in DMSO at 0.47 mg mL$^{-1}$ (1 eq.), 4.65 mg mL$^{-1}$ (10 eq.), 23.25 mg mL$^{-1}$ (50 eq.), and 46.50 mg mL$^{-1}$ (100 eq.). Aliquots of T14 and PY12 were mixed 1:1 (v/v) resulting in the following mixtures: 1:5, 1:3, 1:1, 1:3, and 1:5 eq./eq. (T14:PY12). Subsequently, 1.5 µL of T14 and PY12 solutions was dispensed row-by-row on a G3 dendrimer glass slide A (spot diameter: 2.83 mm, spot pitch: 1.67 mm) and 1.5 µL of each amine solution was dispensed in four replicates column-by-column on a G3 dendrimer glass slide B (spot diameter: 2.83 mm, spot pitch: 1.67 mm). Slides A and B were sandwiched in an 3D-printed sandwiching adapter within a distance of 1 mm. On-chip UV–Vis absorbance of each spot was measured at 365 nm at 5-min intervals over a period of 70 h using a BioTek Synergy H1 UV–Vis plate reader set to 26 °C.

**Cell culture**. Human cervical carcinoma HeLa cells (ATCC® CCL-2™) and human embryonic kidney (293T cells (ATCC® CRL3216™) provided by the Institute of Biological and Chemical Systems (IBCS) at Karlsruhe Institute of Technology (KIT) were cultured in Dulbecco's Modified Eagle Medium (Life Technologies) supplemented with 10% v/v fetal bovine serum (PAA Laboratories) and 1% v/v penicillin–streptomycin (Life Technologies) in a humidified incubator at 37 °C under 5% CO$_2$. Cells were washed with phosphate buffered saline (Life Technologies) and detached with 0.25% trypsin/EDTA solution (Life Technologies). Jurkat T lymphocytes (ATCC® TIB-152™) provided by IBCS were maintained in RPMI-1640 cell culture medium (Life Technologies) supplemented with 10% heat-inactivated fetal bovine serum (PAA Laboratories). Cells were passaged every 2–3 days. Cells were counted using the trypan blue staining (Life Technologies) method in a cell counting chamber (Life Technologies) and analyzed by a Countess Automated Cell Counter (Life Technologies). The cells were centrifuged at 1200×*g* for 3 min at room temperature and the cell pellet was resuspended in freshly prepared medium. Cell suspensions containing 50,000 cells mL$^{-1}$ were prepared and 3 µL was dispensed onto each spot of a dendrimeric-modified slide (spot diameter: 2.83 mm, spot pitch: 1.67 mm) using a non-contact liquid dispenser (I-DOT, Dispendix). The slide was placed into a Petri dish fitted with sterile tissue (Clean and Clever) soaked with 7 mL PBS to prevent evaporation of the droplets and incubated in a humidified incubator for 24 h at 37 °C under 5% CO$_2$. The cells were then stained by adding 1 µL Hoechst 33342 staining solution (1:900 v/v (10 mg mL$^{-1}$, Invitrogen) and 1:1350 v/v propidium iodide (1.00 mg mL$^{-1}$, Invitrogen).

**Image acquisition and analysis**. Fluorescence images were obtained using a BZ9000 (Keyence) fluorescence microscope. The exposure times were set and kept the same for all experiments and repetitions.

*Objectives*: Nikon ×10 PlanApo NA 0.45/4.00 mm; Nikon ×20 PlanFluor ELWD DM 0.45/8.10 mm; Nikon ×40 PlanFluor ELWD DM 0.60/3.70 mm

Resolution: 8-bit

Format: 1360 × 1024 px

Light source: mercury vapor lamp

Cells were counted by adjusting the threshold of the 8-bit images and then using the Analyze Particle function in ImageJ. Cell viability was calculated as the live cell: total cell ratio. All datasets were depicted as mean ± standard deviation. At least three replicates per cell line and per slide were tested for each viability screening.

**Software**. *Data collection*

Water contact angle: Krüss Advance 1.6.2.0; Krüss GmbH
AFM: MultiMode 8-HR; Bruker Corporation
ToF-SIMS: Surfacelab 7.0.106074; IONTOF GmbH
MS: flexControl 4.0.35; Bruker Corporation
IR: Opus 7.8; Bruker Corporation
UV–Vis: UV WinLab 6.0.4; PerkinElmer, Inc./Gen5; BioTek Instruments, Inc.
Microscopy: BZ-II Viewer 1.5.0.0; Keyence Corporation

*Data analysis*:

Water contact angle: Krüss Advance 1.6.2.0; Krüss GmbH
AFM: NanoScope Analysis 1.50 (build R3.119069); Bruker Corporation
ToF-SIMS: Surfacelab 7.0.106074; IONTOF GmbH
MS: flexAnalysis 4.0.14; Bruker Corporation
IR: Opus 7.8; Bruker Corporation
UV–Vis: Office 264 ProPlus Version 1908 Build 11929.20648; Microsoft/OriginPro 2019b Build 9.6.5.169; OriginLab Corporation
Microscopy: BZ-II Analyzer 2.1; Keyence Corporation/Office 264 ProPlus Version 1908 Build 11929.20648; Microsoft/OriginPro 2019b Build 9.6.5.169; OriginLab Corporation

**Reporting summary**. Further information on research design is available in the Nature Research Reporting Summary linked to this article.

## Data availability

The data that support the findings of this study are available from the corresponding author upon reasonable request. Source data are provided with this paper.

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

## Acknowledgements
This research was supported by ERC Starting Grant (DropCellArray, 337077), ERC Proof-of-Concept Grant (CellScreenChip, 680913), Baden-Württemberg Ministry of Science, Research and Arts (DrugMicroarray, 7533-7-11.10-7) and Helmholtz Association. The authors would like to thank Dr. Michael Hirtz from INT (KIT, Institute of Nanotechnology) for providing training and access to atomic force microscopy. ToF-SIMS studies supported by the Laboratory for Microscopy and Spectroscopy of the Karlsruhe Nano Micro Facility (KNMF) are acknowledged.

## Author contributions
P.A.L. conceived the idea of combining miniaturized chemistry, analytics and biology and elaborated it together with M.B.; M.B. proposed the initial idea of the dendrimeric surface modification and elaborated it together with P.A.L.; M.B. designed and constructed the devices, designed the experimental setup, performed the experiments, analyzed, and interpreted the data; A.A. and M.H. performed MALDI-TOF MS characterization and interpreted the data together with M.B.; A.W. performed ToF-SIMS characterization; S.H. performed IR spectroscopy; M.B. and P.A.L. wrote the manuscript with input from all authors.

## Funding

## Competing interests
P.A.L. is a share holder in Aquarray GmbH that develops miniaturized technologies for cell screenings. The remaining authors declare no competing interests.
