## [Peer Review File · Nature Communications]

Reviewers' Comments:

Reviewer #1:

Remarks to the Author:

The authors have demonstrated their chemBIOS platform that combines on-chip chemical synthesis, characterization and biological screening using dendrimer-based surface patterning. This enables the generation of high-density nanodroplet arrays for both organic and aqueous liquids.

Major comments

1. The authors have demonstrated the dendrimer-based surface patterning method on the developed chemBIOS platform (Nat. Comm. 2019, 10, 2879) to increase in functional group density on chip. Please emphasize the originality compared to the previous work. The authors have also applied this approach to three applications; On-chip MALDI-TOF mass spectrometry, On-chip IR spectroscopy and On-chip UV-Vis spectroscopy. Please also describe the originality of the current work across these applications.

2. With respect to the three applications, what are the core advantages that this method provides, particularly compared to other high-throughput platforms?

3. There are examples of dendrimer-based chips and other surface-based approaches. What core unique features are addressed herein?

4. The authors describe effectiveness for biological compatibility. However, the work falls short in actually demonstrating that the system can be used for screening. Just because ~97% cell viability is retained does not indicate that the cells would be functional in a phenotypic screen.

Minor comments

1. The indications of "G0" and "G4" in Figure 2 are not clear.

2. Stainless steel plates were also prepared in Figure 4d. It would be better to emphasize the universal versatility of the chemBIOS platform. This could be achieved by adding one or more applications using such a stainless steel plate.

3. An actual image (photo) of the microplate could be added to Supplementary Figure 8 to help readers visualize the actual system.

4. How did the authors estimate a limit of detection in details (see p. 11, line 198)?

Reviewer #2:

Remarks to the Author:

This is an outstanding piece of scientific and technological work on a high throughput screening platform that combines synthesis of compounds, their analytical and structural characterisation using a broad combination of relevant and well-developed techniques and evaluation of cellular response to a broad compound library. Some of the individual steps in the drug development process used in this work have been published (by the authors of this manuscript), but the big challenge and novelty is the integration of each step in a novel, impressively clever technology. This is the first publication I have seen combining all stages of the evaluation of drug candidates up to the in vitro cell response in a single system approach with minimum amount of drug candidate material and ultrahigh sensitivity recording.

This manuscript is ready for publication in my view. The manuscript is well written, the figures, although complex in terms of the wealth of information, adequate and the overall presentation and discussion quality up to the standard of this Nature journal, in my view.

Response to reviewer's comments

Reviewer #1:

Comment: *The authors have demonstrated their chemBIOS platform that combines on-chip chemical synthesis, characterization and biological screening using dendrimer-based surface patterning. This enables the generation of high-density nanodroplet arrays for both organic and aqueous liquids.*

Response: We thank the reviewer for highlighting the strength of our work and for the important comments that helped us to improve the manuscript. We respond to each comment separately below.

Comment: *Major comments*

1. The authors have demonstrated the dendrimer-based surface patterning method on the developed chemBIOS platform (Nat. Comm. 2019, 10, 2879) to increase in functional group density on chip. Please emphasize the originality compared to the previous work. The authors have also applied this approach to three applications; On-chip MALDI-TOF mass spectrometry, On-chip IR spectroscopy and On-chip UV-Vis spectroscopy. Please also describe the originality of the current work across these applications.

Response: One of the core features of the demonstrated dendrimer-based surface patterning is the possibility to combine chemistry, characterization and biology on a single slide. The demonstrated surface modification is a fast and robust method to manipulate the physicochemical properties of the surface (e.g. wettability) in a controlled manner. We hypothesized that an increase in density of the surface functionalization would correlate with the physical material properties. In this work, we were able to prove this hypothesis by different surface characterization techniques such as contact angle measurements, atomic force microscopy and secondary ion mass spectrometry. The surface-grafted dendrimers could be synthesized layer by layer, thereby properly increasing the density of functional groups immobilized to the surface. Fast, robust, high-resolution and controllable (geometry, position etc.) photochemical omniphilic-omniphobic surface patterning resulted in a high contrast of wettability between omniphilic spots and surrounding omniphobic barriers that enabled us to generate droplet arrays with both low and high surface tension liquids on the same substrate. This opened the possibility to use the omniphilic-omniphobic dendrimer-based patterns to perform on-chip solution-based synthesis (organic and aqueous solvents), on-chip characterization (MALDI-MS, IR or UV-Vis) as well as biological screenings (aqueous media) on the same slide in an iterative cyclic workflow.

We added the following text to the discussion and figure to the Supplementary Information:

"The dendrimer-based omniphilic-omniphobic platform enabled generation of nanodroplet arrays of both low-surface tension liquids (organic solvents) and high-surface tension liquids (aqueous solutions) on the same substrate. This opened the possibility to combine on-chip synthesis, on-chip characterization and on-chip biological screening in an iterative cyclic workflow (Supplementary Fig. 12)."

Supplementary Figure 12 | chemBIOS workflow. Schematic describing the chemBIOS workflow using a dendrimer-modified, omniphilic-omniphobic patterned slide that enables the handling of both low-surface tension liquids (organic solvents) and high-surface tension liquids (aqueous solutions / cell suspension) on the same substrate. On-chip synthesis, on-chip characterization and on-chip screening can be performed using one slide without the need of additional transfer steps.

On-chip characterization is a major challenge arising from the need for miniaturization and parallelization of chemical synthesis. Existing analytical techniques have been developed to be compatible with standard chemistry which is done in flasks requiring relatively large volumes and amounts of substances. Thus, most characterization technologies are not compatible with on-chip high-throughput character, low volumes and low concentrations. However, compound characterization and reaction monitoring are essential in drug discovery to obtain as much information as possible on the investigated substances. An important feature of the demonstrated dendrimer-based surface modification is that the covalently grafted dendrimer layer does not interfere with the optical properties of standard glass slides. The combination of perfect optical properties, miniaturization, parallelization and high-resolution compartmentalization (by manipulating high surface density of functional groups in a controllable manner) enabled for the first time online, in situ, high-throughput UV-Vis reaction monitoring in the droplet array format which is important to monitor the progress of reactions and optimize reaction conditions. Another originality of the current work was the successful modification and patterning of indium-tin oxide (ITO) coated glass slides that does not interfere with conductivity which is crucial for MALDI-TOF MS characterization. Highly dense omniphilic-omniphobic compartmentalization achieved by the dendrimer-based surface patterning enabled ultra-high-throughput characterization (50,400 spots per MALDI target plate). We could show extremely sensitive characterization by MALDI-TOF MS in the attomole range per spot. The geometry and spot size of the dendrimer-based platform could be easily varied by using different photomasks in our demonstrated surface modification method. We observed an enhanced sensitivity of the characterization method by varying the spot size (smaller spots = enhanced sensitivity). Additionally, IR-reflective properties of the ITO coating enabled IR imaging that is originally known from histological investigations of tissue samples. Thus, we developed a method to merge these IR-reflective properties with surface compartmentalization to create a novel system for high-throughput characterization of organic compounds in the array format by on-chip IR microscopy and spectroscopy. Surprisingly, the dendrimer-based surface modification of the ITO layer did not interfere with the IR-reflective properties of the ITO coating which is crucial for performing non-invasive / non-destructive Fourier Transform Infrared Reflection-Absorption Spectroscopy (FT-IRRAS).

Thus, the dendrimer-based platform combines different characterization techniques with each other in one platform to generate a robust novel system that can be used in combination with miniaturized and parallelized organic synthesis. The dendrimer-based

omniphilic-omniphobic patterned conductive ITO coating enables extremely sensitive MALDI-TOF MS characterization that can be combined with (fluorescence) microscopy and optical spectroscopy such as UV-Vis (transparent in the UV-Vis region) and IR (reflective in the IR region). There is no need for any additional transfer steps in the process making the dendrimer-based chemBIOS workflow also faster than the previous methods.

Comment: 2. *With respect to the three applications, what are the core advantages that this method provides, particularly compared to other high-throughput platforms?*

Response: Most established high-throughput platforms rely on a specific readout, e.g. microscopy (microtiter plates), or spectroscopy (plate readers, vials, metal substrates), since each specific readout technique has their own demands on infrastructure (e.g. plates, flasks, bottles, capillaries, syringes etc.) and material properties (e.g. chemical resistance, optical properties, conductivity etc.). As a result, either fewer different characterization techniques are performed in total in large screenings or a large number of transfer steps between individual platforms are required that cause high consumable consumption and prolong the entire work. The dendrimer-based surface modification, which we described in this manuscript, combines excellent optical properties (UV-Vis transparent and IR reflective), chemical resistance (glass substrate compatible with organic solvents), mechanical stability and conductivity and, thus, enables to combine different characterization methods (for chemical and biological readout), such as on-chip mass spectrometry, on-chip spectroscopy and on-chip (fluorescence) microscopy in one unified platform. We believe such a unification is crucial for advancing high-throughput chemical and biological screenings.

In addition to the previous response, we would like to summarize the core advantages of the three characterization methods that are combined in the dendrimer-based chemBIOS platform:

1. MALDI-TOF MS
 - On-chip / online characterization without additional transfer of all samples to a suitable target platform
 - o Straight-forward connection between on-chip synthesis and on-chip characterization
 - Individual on-chip processing of each sample with matrix solution by automated non-contact liquid dispensing
 - Omniphilic-omniphobic surface patterning enables dense compartmentalization and, thus, increase of the throughput (more samples per area compared to conventional MALDI target plates)
 - o Compartmentalization enables position-coded handling and characterization of individual samples
 - Spot geometry / diameter could be controlled by the photochemical surface patterning (higher amount of substance on a smaller spot area resulted in higher sensitivity)
2. IR spectroscopy
 - On-chip / online characterization
 - o Straight-forward connection between on-chip synthesis and on-chip characterization
 - IR-reflective properties of the platform enable non-invasive / non-destructive characterization
 - Dense compartmentalization → increase the throughput per platform
 - Controllable spot geometry → sensitivity
3. UV-Vis spectroscopy

- On-chip / online characterization
 - o Straight-forward connection between on-chip synthesis and on-chip characterization
- On-chip, in situ reaction monitoring
- High chemical resistance of the glass platform (organic solvents incompatible with polystyrene microtiter plates)

We added the following text to the discussion:

“The dendrimer-based surface modification combines compartmentalization, excellent optical properties (UV-Vis transparent and IR reflective), chemical resistance (glass substrate compatible with organic solvents) and conductive material properties and, thus, makes it possible to combine important characterization methods (for chemical and biological readouts), such as on-chip high sensitive mass spectrometry, on-chip spectroscopy and on-chip (optical or potentially electron) microscopy in one multifunctional platform.”

Comment: 3. There are examples of dendrimer-based chips and other surface-based approaches. What core unique features are addressed herein?

Response: The most commonly used type of dendrimers are poly(amidoamine) (PAMAM) dendrimers. Dendrimers are mainly synthesized in bulk or purchased and then immobilized to a substrate (e.g. silicon wafer, glass slide or polymer) on which they act as anchor structures for further chemical or biochemical applications. Due to multiple functional groups, dendrimers show large potential for multiple detection and binding sites, which is especially important for solid-phase synthesis and surface-based biochemical assays (e.g. protein detection, interaction studies etc.) to increase the yield or the detection feedback.

Structuring of a surface by spatially immobilizing dendrimers was done, for example, by microcontact printing or dip-pen nanolithography methods.¹ We take a different path to pattern dendrimer-based surfaces by photo-functionalizing the surface-grafted dendrimers themselves. Here we developed a poly(thioether) dendrimer-synthesis which could be performed directly on-chip to graft the dendrimers layer by layer homogeneously on the surface. The iterative 2-step reaction cycle based on a thiol-ene click reaction followed by an esterification enabled us to synthesize the dendrimer in a controllable manner. We showed, for example, that we can control the thickness and surface roughness of the on-chip grafted dendrimer layer by synthesizing a specific dendrimer generation. The covalently grafted dendrimer layer does not interfere with the optical properties of the glass slide or with the conductive properties of ITO-coated slides which is surprising and crucial for further characterization and readout applications. The core unique feature is, however, the very fast, robust and high-resolution possibility to pattern the reactive dendrimer-based surface by photochemical thiol-ene click reaction subsequently through a customizable photomask. This enabled compartmentalization of the surface for further chemical, analytical and biological applications. This is only possible due to the photo-responsive chemical nature of the alkene-functionalized poly(thioether) dendrimers. Large substrates can be patterned in a single step making this approach compatible with high-throughput. Thus, the developed method is applicable to a much broader community and could open new possibilities in chemical synthesis, biochemical assays, biological studies and material science.

¹ (a) Arrington, D. et al. *Langmuir* **21**, 18, 7788-7791 (2002); (b) Li, H. et al. *Nano Letters* **4**, 2, 347-349 (2002); (c) Salazar, R.B. et al. *small* **11**, 2, 1274-1282 (2006).

Comment: 4. The authors describe effectiveness for biological compatibility. However, the work falls short in actually demonstrating that the system can be used for screening. Just because ~97% cell viability is retained does not indicate that the cells would be functional in a phenotypic screen.

Response: In the submitted manuscript we have shown cell culture of adherent and suspension cell lines followed by cell staining with Hoechst 33342 and propidium iodide and evaluated the cell viability by fluorescence microscopy and morphology. This demonstrated that the surface is not toxic and compatible with cell culturing. However, we agree with the reviewer that the surface could also have other effects on the cells and could, for example, influence the cellular cytoskeleton which is a crucial parameter for more phenotypic screenings. Thus, we performed several additional experiments. First, we performed an actin staining of HeLa-CCL2 cells cultured on the dendrimer surface and compared to the traditional cell culture Petri dish as control. We found no difference in the cell morphology (cytoskeletal distribution) or average cellular area in between these two surfaces (Figure below).

Fig. Peripheral actin staining of HeLa-CCL2 cells cultured on the dendrimer surface and compared with Petri dish. Microscope images of HeLa-CCL2 cells grown for 24 hours on (a) Petri dish and (b) poly(thioether) dendrimer surface. Actin and nuclei were stained by Phalloidin-Atto dye and Hoechst 33342 stain, respectively. Scale bar = 100 μm . (c) Graph showing area of HeLa-CCL2 cells grown on both surfaces, as marked by actin staining. Data represented as mean \pm SEM. n = 20 cells per group.

Next, we investigated the metabolism of HeLa cells cultured on the dendrimer-based chemBIOS platform and in a microplate. Cells were seeded in the same density per area (480 cells mm^{-2}) to thioglycerol-modified G3 spots (spot diameter: 2.83 mm) of the chemBIOS platform and to wells of a 384-well microplate. The cells were cultured for 24h followed by incubating the cells with the cell membrane permeable tetrazolium salt WST-8 (Cell Counting Kit-8; ABP Biosciences) for 1.5h. WST-8 is metabolized and reduced by dehydrogenase activity in the cells resulting in the formation of a yellow-color formazan dye which is soluble in the cell culture media and detectable by absorbance measurements (OD 450 nm). The amount of metabolized WST-8 is directly proportional to the number of living, proliferating cells and thereby an indication of natural metabolic activity. We measured and compared the absorbance of both supernatants from the chemBIOS platform and the 384-

well MTP by a SpectraMax iD3 plate reader (Molecular Devices). Cells cultured on the dendrimer-modified chemBIOS slide showed a similar, slightly increased metabolic activity in comparison to cells cultured in standard microplates which is an important parameter for more detailed cell screenings of bioactive compounds (Figure below).

Fig. Results of the metabolism study. Cells cultured on a 384-well microplate and a thioglycerol-modified G3 chemBIOS slide (spot diameter: 2.83 mm) were treated with a tetrazolium salt WST-8 to observe and compare metabolic dehydrogenase activity which resulted in the formation of a yellow-color formazan dye. $\lambda_{\max} = 450 \text{ nm}$; data represent mean \pm standard deviation based on five repetitions.

However, cell-biomaterial interactions can be best understood by combining phenotypic assay with transcriptomic outcome of the cells grown onto them. In a separate study, we have cultured HeLa-CCL2 cells on the dendrimer surface and traditional Petri dish (control) and performed mRNA sequencing to look for any transcriptomic alterations. We found no striking differences in the gene expressions involved in apoptosis or cellular metabolism upon culturing the cells onto our dendrimer surface. This manuscript is under preparation but is out of scope of this paper as it focuses on transcriptomic analysis of cell-surface interactions. The goal of the present submission was to develop a novel platform with manipulated, omniphilic-omniphobic patterned surface properties that enable a broad variety of applications for miniaturized solution-based synthesis with corresponding reaction monitoring, compound characterization and the possibility of biological applications on the same platform. In another separate follow-up study, we are using the platform for high-throughput cell screenings, but the results go beyond the scope of this work.

We revised the manuscript accordingly:

“The viability of three cell lines (HeLa, HEK293T and Jurkat) dispensed to each spot of a thioglycerol-modified G3 slide and cultured for 24 h was evaluated by fluorescence microscopy after life/death staining and by morphology (Supplementary Fig. 11). The viability of each cell line exceeded 97%, demonstrating the compatibility of this platform for cell culturing and opened the possibility for further biological screenings in follow-up studies.”

Comment: *Minor comments*

1. The indications of "G0" and "G4" in Figure 2 are not clear.

Response: We agree with the reviewer. In figure 2, we show the process of manufacturing G3-modified slides, which we used for all applications that we describe in the following (on-chip synthesis, characterization and cell culturing). We added "G0" to figure 2 but (with regards to the limit of available space) we suggest to not add the structure of "G4" dendrimers in this figure. However, we added the chemical structure of G4 dendrimers to the Supplementary Information.

Supplementary Figure 1 | Chemical structure of a surface-grafted G4 dendrimer.

Comment: 2. *Stainless steel plates were also prepared in Figure 4d. It would be better to emphasize the universal versatility of the chemBIOS platform. This could be achieved by adding one or more applications using such a stainless-steel plate.*

Response: We agree with the reviewer and we performed several additional MALDI-TOF MS experiments on the omniphilic-omniphobic patterned stainless steel plate to demonstrate the applicability of MALDI-TOF MS on a broad variety of other substance classes:

- Peptides (tryptic digestion of bovine serum albumin)
- Proteins (myoglobin)
- Carbo hydrates (maltoheptaose)
- Drug compound (Verapamil)

Transparency is crucial for other characterization techniques which is why our focus was on the development of the modified transparent ITO slides.

We revised Figure 4 to emphasize our ITO-related work:

Figure 4 | Indium-tin oxide substrates enable on-chip characterization by MALDI-TOF mass spectrometry and IR spectroscopy. (a) Schematic showing the process of on-chip characterization by MALDI-TOF MS. A compound library generated on a conductive, dendrimer-modified and patterned ITO slide can be processed and co-crystallized with matrix solution prior to MALDI-TOF analysis. (b) MALDI-TOF mass spectra of 2, 0.3 and 0.1 fmol per spot of lipidoid 1. Spot diameter: 500 μm ; spot distance: 250 μm . The MALDI-TOF measurements were performed on patterned ITO glass slides. (c) Bar chart showing signal-to-noise (S/N) ratio obtained from on-chip MS analysis of lipidoid 1 in spots of different sizes. Data represent mean \pm standard deviation based on triplicate experiments. (d) Schematic diagram showing the process of on-chip characterization by IR spectroscopy. Non-invasive, on-chip characterization of a compound library by IR spectroscopy acquired after evaporation of the solvent. (e) On-chip IR imaging of several spots containing different amounts of lipidoid 1 per spot. (f) IR spectrum of lipidoid 1 (94.7 fmol per spot). Spot diameter: 500 μm ; spot distance: 250 μm .

We added the following figure to the Supplementary Information:

Supplementary Figure 8 | MALDI-TOF MS on an omniphilic-omniphobic patterned stainless steel plate.

(a) Photograph of a dendrimer-modified and patterned stainless steel plate of microtiter plate size presenting 50,400 individual droplets. Spot size: 330x330 μm^2 ; spot distance: 60 μm ; solvent: DMSO. Mass spectra measured on patterned stainless steel plate of (b) 25 femtomole myoglobin, (c) 50 femtomole Verapamil, (d) 250 attomole tryptic digested bovine albumin and (e) 1 picomole maltoheptaose.

We added the following text to the manuscript:

“A broad variety of substance classes could be analyzed on the dendrimer-based omniphilic-omniphobic patterned stainless steel plates including peptides, proteins, carbohydrates and small molecule drugs (Supplementary Fig. 8).”

We added the following text to the method section:

“MALDI-TOF spectra of further classes of test substances measured on a patterned stainless-steel plate covering a wide range of molecular size (peptides, proteins, carbohydrates, small drug molecules) were acquired applying the following experimental conditions:

Bovine Serum Albumin tryptic digest: MALDI matrix HCCA; reflector positive operation mode; 6000 laser shots accumulated

Myoglobin (horse): MALDI matrix 2,5-DHAP; linear positive operation mode; 5000 laser shots accumulated

Maltoheptaose: MALDI matrix 2,5-DHB / 1mM NaCl; reflector positive operation mode; 4000 laser shots accumulated

Verapamil: MALDI matrix HCCA; reflector positive operation mode, 4000 laser shots accumulated”

Comment: 3. An actual image (photo) of the microplate could be added to Supplementary Figure 8 to help readers visualize the actual system.

Response: We agree with the reviewer and revised the figure accordingly:

Supplementary Figure 10 | 3D-printed sandwiching adapter for UV-Vis measurements. (a-b) Concept art images of a droplet array trapped between two dendrimer-modified, omniphilic-omniphobic patterned slides. The slides were sandwiched using a 3D-printed adapter. **(c) Photograph of the 3D-printed sandwiching adapter. (d)** Schematically showing the top view of the sandwiching adapter with overlaid 384-well microtiter plate patterns. Blue: omniphilic-omniphobic patterned glass slide.

Comment: 4. How did the authors estimate a limit of detection in details (see p. 11, line 198)?

Response: We analyzed the signal-to-noise ratios of different amounts of substance per area unit. We estimated the limit of detection from spectra of which we could identify and separate signals from the target compound by comparing the corresponding spectra of the

negative control (measured matrix spectrum as reference). We added the following text to the methods:

“S/N ratios of each amount of substance per area unit were determined and the limit of detection was estimated by comparing the sample spectrum with a blank matrix spectrum as a reference.”

Reviewer #2:

Comment: *This is an outstanding piece of scientific and technological work on a high throughput screening platform that combines synthesis of compounds, their analytical and structural characterisation using a broad combination of relevant and well-developed techniques and evaluation of cellular response to a broad compound library. Some of the individual steps in the drug development process used in this work have been published (by the authors of this manuscript), but the big challenge and novelty is the integration of each step in a novel, impressively clever technology. This is the first publication I have seen combining all stages of the evaluation of drug candidates up to the in vitro cell response in a single system approach with minimum amount of drug candidate material and ultrahigh sensitivity recording.*

This manuscript is ready for publication in my view. The manuscript is well written, the figures, although complex in terms of the wealth of information, adequate and the overall presentation and discussion quality up to the standard of this Nature journal, in my view.

Response: We thank the reviewer for highlighting the strengths and novelty of our work and for recommending publication of the manuscript without the need of changes.

Reviewers' Comments:

Reviewer #1:

Remarks to the Author:

The authors' revision is thorough and covers each of my points in detail, and as a result I only have one final comment/suggestion based on their responses to my first two major comments emphasizing the novelty of this work over the prior chemBIOS work published last year (Nat. Commun. 2019, 10, 2879). The edits help define the benefits of the dendrimer-based patterning of the chemBIOS platform later in the text; however, the document would likely benefit from defining this unique aspect of the work more clearly in the introduction. Perhaps, can state in the introduction (potentially by modifying the sentence on p. 3, lines 51-54) that their dendrimer-based surface patterning method "enhanced" or "evolved from" the existing chemBIOS (Nat. Commun. 2019, 10, 2879) platform to "expand its capabilities", thereby providing novel functionality previously not possible with the original platform. Adding such text would clearly define the difference in the work and put a spotlight on how these modifications improve the workflow as shown in the newly added Supplementary Figure 12, a fact that would emphasize how the authors are working to address the deficiency in current analytical methods (p. 3, lines 45-51).

Response to reviewer's comments

Reviewer #1:

Comment: *The authors' revision is thorough and covers each of my points in detail, and as a result I only have one final comment/suggestion based on their responses to my first two major comments emphasizing the novelty of this work over the prior chemBIOS work published last year (Nat. Commun. 2019, 10, 2879). The edits help define the benefits of the dendrimer-based patterning of the chemBIOS platform later in the text; however, the document would likely benefit from defining this unique aspect of the work more clearly in the introduction. Perhaps, can state in the introduction (potentially by modifying the sentence on p. 3, lines 51-54) that their dendrimer-based surface patterning method "enhanced" or "evolved from" the existing chemBIOS (Nat. Commun. 2019, 10, 2879) platform to "expand its capabilities", thereby providing novel functionality previously not possible with the original platform. Adding such text would clearly define the difference in the work and put a spotlight on how these modifications improve the workflow as shown in the newly added Supplementary Figure 12, a fact that would emphasize how the authors are working to address the deficiency in current analytical methods (p. 3, lines 45-51).*

Response: We thank the reviewer for the comment and revised the Introduction accordingly.

"In this work, we develop a dendrimer-based surface patterning method, evolved from the existing chemBIOS platform to expand its capabilities,¹ that can be used to handle high-density nanodroplet arrays of both low (organic solvents) and high (aqueous solutions) surface-tension liquids, thus, enabling a broad range of chemical, analytical and biological applications (chemBIOS) (Fig. 1)."

1. Benz, M., Molla, M.R., Böser, A., Rosenfeld, A. & Levkin, P.A. Marrying chemistry with biology by combining on-chip solution-based combinatorial synthesis and cellular screening. *Nature Communications* **10**, 2879 (2019).